# Talking about Meaning and Loss with Relatives of Persons with Dementia: An Ethnographic Study in a Nursing Home

**DOI:** 10.3390/geriatrics8010023

**Published:** 2023-02-03

**Authors:** Caroline Couprie, Jenny T. van der Steen

**Affiliations:** 1Department of Theology and Religion Studies, Spiritual Care, VU Amsterdam, De Boelelaan 1105, 1081 HV Amsterdam, The Netherlands; 2Pieter van Foreest, Kalfjeslaan 2, 2623 AA Delft, The Netherlands; 3Department of Public Health and Primary Care, Leiden University Medical Center, Hippocratespad 21, Gebouw 3, 2300 RC Leiden, The Netherlands; 4Department of Primary and Community Care and Radboudumc Alzheimer Center, Radboud University Medical Center, 6500 HB Nijmegen, The Netherlands

**Keywords:** dementia, spirituality, meaning, loss, question prompt list, advance care planning, nursing home

## Abstract

Advance care planning (ACP) can help prepare for future losses and decisions to be taken. However, relatives of persons with dementia may wait for healthcare professionals to initiate ACP conversations which may not adequately address their individual information needs. To evaluate inducing and enhancing conversations about meaning and loss, we conducted an ethnographic study on nurse-led ACP conversations using a question prompt list (QPL) on six dementia wards of a nursing home in the Netherlands from January to September 2021. Staff received training in using the QPL, with information and sample questions to inspire relatives to ask their questions, in particular on meaning and loss. Thematic analysis was applied to transcribed interviews and memos of observations. Nursing staff in particular was concerned about having to be available to answer questions continuously. Relatives used the study as an opportunity to get in touch with professionals, and they saw the QPL as an acknowledgement of their needs. There was a mismatch in that staff wished to discuss care goals and complete a care plan, but the relatives wanted to (first) address practical matters. A QPL can be helpful to conversations about meaning and loss, but nursing staff need dedicated time and substantial training. Joint agenda setting before the conversation may help resolve a mismatch in the preferred topics and timing of conversations.

## 1. Introduction

A diagnosis of dementia can trigger relatives to all kinds of questions about the future. Initially, many questions can be answered by healthcare professionals (for example, the doctor, but also the nurse or social worker) when, in the community setting, they convey the news of the diagnosis or provide aftercare. However, there are also questions that may surface later during the disease trajectory [1,2].

Residents in nursing homes experience multiple losses, including loss of contact with the outside world, loss of control and loss of individuality [3]. Nursing home residents are typically in a late phase of life. Issues around meaning and loss are no less relevant after admission to a nursing home, when residents lose their familiar home and the context that supports their identity. Nursing home residents have experienced and need to cope with such loss. Focusing on relationships could support care planning and the further development of a palliative care model for dementia [4,5,6]. When the person with dementia is no longer able to participate in advance care planning (ACP) conversations, relatives voice the needs and preferences of this person. ACP is a process in which residents’ values and preferences regarding care are discussed so that they are aligned before the person is no longer able to formulate them as the disease progresses, after which it continues with relatives.

The disease trajectory in dementia is rather unpredictable, but declining cognition is certain, and timely ACP is recommended [7]. A question prompt list (QPL) (a leaflet or booklet with sample questions, sometimes accompanied by information) enables relatives to better steer ACP conversations towards issues they find relevant at that point, by drawing inspiration from the information and sample questions given.

A study conducted among 174 cancer patients showed that patients ask more relevant questions if they read up on the issues beforehand and are provided with a QPL [8]. Patients have indicated that good communication is an important aspect of good care. Yet, for several reasons, it can be difficult for patients to phrase their questions; the research of Clayton et al. [8] shows that healthcare providers miss hints and cues when patients ask their questions. In addition, patients need guidance and encouragement to ask and formulate questions [8]. A review on cancer patients found that a QPL is effective in communicating with cancer patients. For example, studies show that if the patient receives information that focuses on his or her personal situation, he or she will also feel better psychologically, because the long-term use of a QPL reduces anxiety (in the short term, anxiety can actually increase) [9]. A QPL provides an opportunity to become an informed partner in ACP conversations. They can structure a conversation, that is, the topics and questions can be treated in a structured manner based on the needs of the person and the relative. In this way, the person and relative are informed in advance and can therefore set their agenda and ask more specific questions or steer the conversation towards topics relevant to them. Research shows that caregivers of people with dementia often focus on information related to future care needs, as opposed to caregivers of people with cancer, who often focus on remission. In addition, it appears that caregivers of people with dementia are much less confident in their communication than caregivers of people with cancer [10]. Brazil et al. [11] found that advance care planning helps to reduce this uncertainty. They therefore recommend promoting ACP conversations between relatives and care providers.

For people diagnosed with dementia and their families, QPLs have been developed in several countries [12,13,14,15]. However, research on use or impact of QPLs in practice is scant in the case of dementia. Regarding issues important to persons with dementia, Camacho-Montaño et al. [16] found in a qualitative systematic review that the spiritual needs of persons with dementia are very important, but these domains did not appear in available measurement instruments for quality of life of persons with dementia. One of the QPLs intended specifically for dementia in the last phase of life [7,17] includes a category about meaning and loss. This study focuses on this particular part of the QPL, as many other studies in other settings and populations have also shown that talking about purpose and meaning, despite being considered part of good palliative care, is easily overlooked in conversations with healthcare professionals [16]. The QPL could be particularly helpful in discussing relatively taboo topics, such as perceptions of inevitable decline, and any resulting existential questions relatives may have too. Information and questions about these topics are then also provided and brought to the fore, and can be discussed at the discretion of the relative. Therefore, the QPL investigated in this study is expected to get relatives to talk more about how they experience the disease process of their loved ones and how they interpret it, so that appropriate (future) care can be provided.

This research introduced the dementia QPL in practice. It examines what happens in a context in which doctors are generally in charge of an ACP that is often focused on setting specific advance medical treatment orders [18,19]; nurses and relatives receive a QPL and are asked to talk about meaning and loss. The first objective is to evaluate the induction and enhancement of conversations about meaning and loss in the context of nurse-led ACP conversations in dementia. The second objective is to evaluate the extent to which a QPL that includes a section about meaning and loss can meet the needs of relatives of nursing home residents with dementia and nursing staff.

## 2. Methods

### 2.1. Setting

The study was conducted in a nursing home facility in the Netherlands on 6 dementia care units (‘psychogeriatric’ units; almost all dementia). In the Netherlands ‘elderly care’ medicine is a special medical discipline for long-term care [20,21]. Many nursing homes in the Netherlands employ certified ‘elderly care’ physicians who generally conduct ACP conversations [18,19]. There are few nurses and nurse practitioners, and the nursing staff comprises mostly nurse assistants and aids. Because many residents on dementia care units are no longer involved in ACP conversations and unable to answer interview questions, we involved their relatives. These relatives also make decisions about treatment and care together with the elderly care physician and nursing staff. Upon admission, the relatives are invited for a meeting to become familiar with the nursing home’s routines, and agreements are made. After about 6 or 8 weeks, there is a follow-up consultation in the form of a multidisciplinary consultation that usually results in a care plan which may or may not address specific end-of-life issues.

### 2.2. Design

Initially, a qualitative study design was chosen with repeated open (minimally structured) interviews. We had planned to conduct one interview before the ACP conversation introducing the QPL, and another interview afterwards. We used the ‘consolidated criteria for reporting qualitative studies’ (COREQ) 32-item checklist for the reporting of our study [22]. Actual or planned use of the QPL was not required to participate in the interview in order to enable identification of any impediments to its use. The interviews were, with the consent of the participants, audio recorded.

What stood out, however, was that much valuable data for the research were mentioned after recording the interviews. The conversations outside the interviews were less tense, and therefore much more was elaborated on relevant topics. It was therefore decided to supplement the data from the interviews with newly collected data from observations and fieldnotes. The researcher made extensive notes of her observations and informal conversations. She also described her own behaviour and attitude, and reflected on her presence during the visits. In a later phase (June 2021), the interviews of the planned method (a qualitative interview study) were integrated into an ethnographic study (the duration of the entire study was January 2021 to September 2021). 

#### Description of the Population

The initial aim was to include an entire care team in the study. Professionals from more disciplines were asked to participate in this study, including a physician and nurse assistants. The latter, called an EVV-er (literally, “first responsible nurse assistant”) is responsible for nursing care for the selected resident, and is the first healthcare professional contacted by the main relative. In addition, social workers, psychologists, registered nurses, nurse practitioners and spiritual counsellors were also approached to participate. The participants were invited to participate in person, by email or via a poster.

The healthcare professionals were asked to provide informed consent to participate in the study, which involved offering ACP conversations to the relative, one training session with role play and two interviews with the researcher. The researcher handed out the QPL that combined information and sample questions, along with a separate list of all questions to facilitate choosing questions and making notes. Users can check the box for the question they want to discuss in the conversation. The ethnographer (CO) encouraged relatives to bring the QPL to the conversation. The ethnographer interviewed the participating relatives and explained the purpose of a QPL; she instructed the participating healthcare professionals on how to enter into a dialogue about their consultation with residents and their relatives. The professionals were asked if the relative or the resident, if able, agreed to review and discuss the section on meaning and loss. However, the family and professionals were assured they were free to discuss any part of the QPL. The healthcare professionals agreed with the relative about which topics they wanted to discuss before they started the conversation. All participants received a brief questionnaire to assess some personal details that would describe the sample in study reports.

The fieldwork was carried out by CO, who worked 4 days as a spiritual counsellor in a nursing home in the Netherlands. CO conducted formal interviews with relatives of residents and with professionals, and attended meetings. To avoid undue influence, she had no care relationship with participants who signed up for the study, and the interviewees were explicitly asked which information was reported in confidence and which information could be shared for the study. She also had no hierarchical relationships with the participating professional caregivers.

### 2.3. Analysis

The recorded open interviews were transcribed verbatim. These data were analysed in order to address the two objectives of the research. The analysis was based on thematic analysis, which included (1) becoming familiar with the data, (2) coding, (3) finding themes, (4), revising the themes, (5) defining and naming the themes, and (6) reporting the themes [23]. To increase the reliability of the analysis, another researcher (JTS) independently coded a few interviews and discussed the results with CO [24]. After a few months, the findings were sent to the participants for a member check. The participants were given the opportunity to comment on these findings by phone or in an extra conversation. All participants were contacted by phone for a member check; none of them requested an additional conversation.

The transcripts of 30 detailed descriptions of meetings, observations in departments, communication between participants and short conversations were analysed. The average time span of observations was 3 h. A code tree (Table 1) was set up to identify patterns. The transcripts were coded using open codes. Next, axial coding was carried out, and these data where combined. The interviews (Table 2) and the observations were compared with each other and similarities and differences between topics were analysed. Overlapping codes were merged. Data saturation was achieved after 5 months of the 7-month data collection; the identified patterns were checked with the descriptions of the last 2 months.

## 3. Results

### 3.1. Experiences with the ACP Intervention

We identified six themes in the data of the interviews and observations: (1) atmosphere in which there is room to identify and address needs, (2) ACP: reactive and proactive care, (3) misunderstanding and conflict, (4) acceptance of the QPL, (5) loss, and (6) meaningful daily activities.

### 3.2. Context at the Time of the Fieldwork

Coronavirus measures meant that relatives were no longer invited to multidisciplinary meetings unless something escalated around the care of a resident. The routine process was not always carried out as planned; sometimes the conversations were conducted online, and sometimes they took place without the relative. Consequently, the relative had very few contact moments with healthcare professionals. Taking part in the study stimulated the relatives to make several attempts to get in touch with a healthcare professional. As soon as they succeeded, this moment of contact was experienced as positive. Another consequence of the coronavirus measures was that the training had to be given online.

### 3.3. Participation of Staff

There were three scheduled training opportunities, none of which the participants attended. To be able to follow up on this research, it was decided to make a short recording with the input of external experienced trainers (two nurse practitioners and a researcher on ACP in the nursing home, and CO and JTS attending as well), and with an explanation of the background of the research. Along with this recording, the researcher visited the participants individually to provide training. The participants indicated that they liked the fact that individual training was given specifically given their position. 

During this individual training, the participants asked many questions about how the QPL should be introduced into the conversation, although all professionals were already familiar with the concept of ACP through training from the organization or from their previous education.

There were 12 applications from staff members who did start the study but who dropped out later. The reasons for dropping out were not wanting training (even though they were informed in advance), not daring to speak about sensitive topics with relatives, short-term or long-term illness and finding a new job. Some of these staff members do appear in the field notes on observations as part of the ethnographic study. The characteristics of the interviewees are shown in Table 3.

### 3.4. Atmosphere in Which There Is Room to Identify and Address Needs

The interviews and observations showed that the coronavirus-induced measures had a major impact on the ACP process in terms of opportunity for relatives to connect with nursing staff. The fact that relatives could not join the multidisciplinary consultation was most often mentioned as a shortcoming. All interviewed relatives missed the care consultations with the staff. Some staff were aware that relatives may have missed these moments, but they saw the advantage that the multidisciplinary consultation proceeded much more efficiently without relatives.

The term workload came up various times. On the one hand, more administration tasks were expected from the employees and on the other hand, there was a shortage of permanently employed staff. More employees were being employed on a flexible basis, but they were exempt from administrative duties. This implied a higher administrative workload for those with permanent contracts, in addition to increased tasks of having to coordinate the work of employees with flexible contracts. The following quote shows an employee expressing frustration at work pressure:
*‘I’m so tired of this, it’s all written down but nobody reads it. That’s a team thing as well. But there you go again, when you’re understaffed, when you are with a flex worker. Don’t you think, guys, the new care plans are ready and signed! We are ready to go for another six months.’*

Further, relatives felt it was difficult to get in touch with the right staff member. Relatives are expected to have conversations with their personal ‘EVV’er’ only during contact moments such as a multidisciplinary consultation. By contrast, relatives felt the need to speak to someone on the ward when they visited, rather than having to wait for a scheduled conversation.

### 3.5. Advance Care Planning: Proactive and Reactive Care

Only the highly skilled employees were at ease with ACP. Staff willingness to commit to ACP also influenced the ACP taking place:
*‘So only when someone shows real problem behaviour, then you start asking, well what was their life like and then we dig a little deeper and we think, oh, let’s try this, but that’s not standard, while for many people it might have been a very natural aspect of their life before they were admitted.’ (Physician, woman). A little later she indicates: ‘But it’s not like you just have room in your schedule to sit down and chat with someone for an hour about the future; but it’s something that plays a role in all your decisions.’*

Time also emerged as an important factor in many observations, and it influenced whether proactive or reactive care was provided. A lack of time also meant that the response was mainly reactive:
*‘In a multidisciplinary meeting (during which care goals for a resident are discussed) it was announced that there was little time for the consultation, so only the main points could be discussed’, ‘What was striking was that the employees tended to talk mainly about practical matters that need to be arranged urgently’ (Fieldnote).*

Nevertheless, one employee saw that talking about ACP based on the QPL can also save time because of the proactive approach to care:
*‘And if you record everything properly. By entering in a resident’s plan or care profile, because you’ve asked the right questions, that saves a lot of work. I do think that’s tricky, because people will think, oh, now I have to do something extra, but actually it’s not something extra’ (Nurse, woman).*

### 3.6. Misunderstanding and Conflict

None of the relatives were satisfied with the care provided in the nursing home. This dissatisfaction was expressed in all their interviews, which resulted in the vast majority of codes being classified as conflict and misunderstanding, a theme that also covers a great diversity of codes. Relatives reported conflicts during the interviews, and several conflicts were observed.

The observed conflicts were largely one-off conflicts, but the conflicts from the interviews were ongoing and of longer duration:
*’Yes, yes we are very angry about that. And now I have to sort it out with the nutritional assistant? That is his job. He was supposed to do that. And he was to talk to her and inform me. Well and then he went on holiday. So I think, so we’re six weeks on and actually nothing happened.’ (Relative, woman)*
*‘One care worker gave the example that she had a relative who came round once every fortnight on a Friday evening and expected nurse assistant to be available because she came in so infrequently’ (Observation)*

Because of these conflicts, the role of the interviewer sometimes shifted to mediator and sometimes to spiritual counsellor. Some interviews were stopped and resumed later because it was decided to focus on the conflict first. In other interviews, the opposite choice was made, that is, the interview was completed before attention was paid to the conflict. The main reason why a conflict escalated was that relatives no longer knew how to get in touch with the right person. In most conflicts, relatives felt that information was not properly communicated; other conflicts concerned staff attitudes. Relatives in particular wanted to be informed more often of changes in care provision. Relatives wanted to talk about practical matters such as compressions stockings, food, and heating. Staff believed that care planning conversations are not intended to discuss these topics. 

### 3.7. Use, Acceptance, and Concerns around the QPL

Most relatives and professionals were positive about the QPL. In particular, the gesture of the QPL being there to enable better conversations was experienced as very positive. Because of this signal, people felt heard and seen, and the openness made people experience that difficult topics can also be discussed. In addition, the relatives also saw it as an acknowledgement of the difficult process they go through. What was experienced as very positive was the information provided in the QPL. It triggered some relatives to consider and to contemplate topics that were presented. Others indicated that they had been searching for information for a long time and that they would have liked to see this QPL sooner. Some would have liked the QPL to be even more extensive with specific information about the particular nursing home in question. Finally, the QPL was also seen as a good tool to prepare for the conversations:
*‘I had prepared some questions using the interview guide, like: I want somewhat meaningful daytime activity for her, and to what extent a day is meaningful’ (Relative, woman)*

However, during the conversations with healthcare professionals, relatives did not refer to or bring the QPL into the conversation. Relatives felt more confident going into the conversation because they were prepared. One participant indicated that he often felt overwhelmed by information and decisions during conversations with healthcare professionals. The QPL helped him to prepare for these conversations so he had more time to think about a decision. 

There was also criticism of the QPL. Staff members in particular saw negative effects in the use of the QPL. One of their major concerns was that it would increase their workload:
*‘If a family member already has many questions and they come up at every opportunity, like mushrooms popping up all over the place, and you have to have a conversation every time, then I think in that case, you can say, well we’ll schedule an hour sometime and then we’ll sit down and we’ll send them in advance, something like that, I could imagine that.’ (Nurse assistant, woman)*

It is a large extensive product, and it gives relatives even more tools to ask questions, on top of staff’s perceived lack of time to answer any questions properly. Some employees were convinced that the organization was not ready for a QPL; they felt that staff were not yet sufficiently attuned to each other, and that adequate implementation of a QPL would not be possible. 

Regarding loss, although the intention of the research was to cover the section devoted to the two topics of meaning and loss, none of the interviewees raised loss as a topic of conversation. When the researcher introduced the subject of loss into the interview, relatives generally reported the loss experiences that they themselves experience when they see that someone is deteriorating due to dementia:
*‘And then when you are confronted with it, of course it is still on your mind and you have your own feelings about it, because it, it violates, it goes against your feeling. You don’t want to move someone, you don’t want them to be in pain, you also don’t want them to no longer recognize you, to not understand you; that is all part of it of course, you also don’t want deal with a person’s incontinence.... (Relative, woman)’*

The loss experiences of residents also confronted staff with their own losses. On one occasion it was decided not to continue an interview about the QPL because they themselves shared their loss experiences. One nurse observed that residents experience loss when they come in:
*‘People have lost so much when they come to us, social contacts but also in terms of functioning, (...) and losses in activities as well, people who used to be able to do needlework but no longer can. I think we often know that this is the case, but very often we just don’t handle, don’t adequately plan for it or something.’ (Nurse assistant, woman).’*

There was one exception in which a relative did not speak about her own loss, but spoke about which losses she most recognized as the resident’s loss:
*‘And my mother has difficulty accepting that someone else has to clean it up for her’ (Relative 3, woman).*

### 3.8. Meaningful Daily Activities

This section observed the expressions of residents with dementia, considering existential meaning. One relative commented on meaning:
*‘Meaning for her? No, she does not have that!’ (Relative 1, woman).*

When describing meaning as meaning found in daily activities, in ‘What makes a day meaningful?’ there were most comments (3 professionals and 3 relatives) about missing meaningful activities on the ward:
*‘My mother doesn’t do anything anymore and it’s like she is sleeping all day.’ (Relative 2, woman).*

What were perceived as meaningful daily activities were, for example: ‘letting her set the table’, ‘picking out clothes from fashion magazines’ and ‘recycling waste’. For one relative, the QPL resulted in her starting a conversation about meaningful daily activities. She was very enthusiastic about the outcome of this conversation. She said she would never have started this conversation on her own, without the QPL.

One participant indicated that she also sees participating in this research as meaningful:
*‘Well, because I really do want to stay involved with her and that I also want to do something, sort of be meaningful in the future for other people who end up in this situation (Relative 1, woman)’.*

## 4. Discussion

The first objective of this ethnographic study was to evaluate the induction and enhancement of conversations about meaning and loss in the context of nurse-led ACP conversations in dementia. It was difficult for professionals to free up time for the training, and they did not feel confident enough to lead the conversation. The relatives however, saw the QPL as an acknowledgement of their needs for information and of their need to connect with the professionals about care. However, there was a striking mismatch between the preferred focus of the conversation of the relatives and that of the professionals. The professionals wanted to talk about care goals and the completion of the care plan, and such a focus was not helpful for conversations about meaning and loss. By contrast, the relative wished to (first) address practical matters such as the environment in the nursing home, laundry, and food. The topics that relatives wished to discuss were not part of the QPL, and it was not necessary for professionals to inspire them to talk about these types of topics. This shows that professionals and relatives also had different needs regarding the timing of discussion of the contents of the QPL. The relatives preferred brief contact moments when they visited the nursing home, whereas the healthcare professionals preferred scheduled conversations such as a multidisciplinary consultations. In order to facilitate initiation of ACP conversations, it is important to first discuss with the relatives whether they have any points of dissatisfaction, to recognize their feelings and if possible to resolve the issues or plan another conversation about these points before ACP conversations.

Limited contact with family members and the inability to participate in family gatherings due to coronavirus measures may have changed the way family members interact with professionals around ACP. This may also have influenced their loss experiences.

An explanation for the need of relatives to discuss practical matters (that are not included in the QPL) could be that there were no scheduled moments to discuss these matters. Some relatives would prefer information that was more specific for the particular nursing home. Some also felt that the information provided in the QPL should have been shared much earlier, before their loved one moved to the nursing home. It was remarkable that all relatives who agreed to participate were dissatisfied with the care provided (at that time), and all nursing staff who agreed to participate indicated they were already familiar with ACP.

When comparing the results to research by Clayton et al. [25] who found that both patients diagnosed with cancer and doctors experienced a QPL as a valuable addition to their conversations, our results show that there is a difference in the welcoming of the QPL between professionals and relatives. Perhaps, unlike nursing staff, staff members such as doctors and psychologists experience fewer barriers to using the QPL, because they do not walk around the ward and can therefore allocate dedicated time for these conversations. This may explain why nursing staff were concerned about adding to their workload by using the QPL.

Another study [26] used a one-page checklist with common questions, and showed that a QPL can be helpful; however, remarkably, none of the patients actually handed the QPL to the oncologist, even though this was stimulated beforehand. This shyness was also seen in this study, while the professionals did ask many questions during the instruction about how the QPL should be used. Relatives had also taken in the information from the QPL and sometimes formulated questions themselves, but they did not refer to the QPL during the conversations. 

Walczak et al. [27] found that a QPL is valued by people with high information requirements. They distinguish people with a high information requirements from a group with a very high information requirements. For the latter, the QPL did not add value because they had already asked or looked up the information and questions themselves. Something similar was observed in this study. People with a high requirement for information (those whose searches are more goal oriented, and whose information needs are more focused on what they do not know) also look for other sources to obtain it, such as a social worker.

Our findings indicate that the term daily meaning was better suited to the relatives and professionals than existential meaning. Existing sources of significance, such as the church, have disappeared for many in a secularized country such as the Netherlands, and according to Baumeister and Muraven [28], this has two important consequences. First, the term ‘meaning’ is becoming more commonplace, and second, important choices people make are becoming much more of an individual responsibility.

Studies have shown that people with dementia can experience a deep understanding of spirituality. Memories of religious and spiritual experiences from their lives can stir this up [29]. In the interviews, there was no mention at all of the religious or spiritual experiences of the residents with dementia. None of the relatives in our study were connected to a religious community, and they may have been unfamiliar with religious or spiritual language. Our study suggests that it can also be difficult for relatives to see existential meaning in people with dementia. Not talking about loss could be a coping strategy to deal with the current situation. Still, further research is needed to explain why relatives did not bring up the topic of loss and existential meaning in conversations, despite being encouraged to do so. The conversations about loss evoked emotions among healthcare professionals as they touched upon personal experiences of loss. Additional research is needed into the extent to which healthcare professionals being open to speaking about personal loss experiences can stimulate ACP conversations with relatives. 

### 4.1. Limitations and Strengths of the Study

Expanding the qualitative interview study design to ethnographic fieldwork resulted in a better understanding of how the ACP conversations were being prepared and conducted, and how the QPL was used or why it was not used in practice. The fact that the ethnographer already had established links within the organization was both a strength and a weakness. An advantage that facilitated reverting to an ethnographic study was that a relationship of trust had already been built, so that staff and relatives dared to express themselves more about their situation. They even offered additional information by making contact and explaining their personal experiences outside the interviews. The disadvantage was that the roles of interviewer and spiritual counsellor were sometimes difficult to separate. In the Netherlands, chaplains operate independently of the multidisciplinary team and do not have to share or document anything that comes to their attention with the care or medical staff. People in the role of chaplain are taught not to take sides, and an oath of confidentiality is taken. In order to distinguish between the role of chaplain and researcher, it was agreed with the participants what may be shared in the research publication. The personal interpretation of the researcher was discussed on the basis of transcripts and situation descriptions. An example of this is that conflict resolution had to be prioritized in day-to-day practice, which at times required delaying the timing of interviews, whether with investigators or for ACP, in order to achieve longer-term goals. Another possible bias is that interviewees might have had an unspoken expectation that the interviewer could help them solve the conflicts they experienced. To mitigate such possible expectation, the interviewer emphasized that these conversations are only for the benefit of the research, and she referred to the managers for conflict resolution as needed.

### 4.2. What This Study Adds: Implications

Most studies investigating the use of a QPL have taken place in a short-stay environment or an ambulatory care setting such as a hospital or general practice, while our study was situated in a nursing home environment, in which nursing staff engage in long-term relationships. In this study, general training was provided for all disciplines and nursing levels. Because there was little interest in group training, we offered individual training, which, however, may not be enough. In addition, when it comes to loss and meaning, it is important to have an eye for personal feelings that can play a role in discussing these topics. 

## 5. Conclusions

Further research is recommended into the differences in QPL use between disciplines, such as the medical and psychological disciplines, spiritual counsellors (for meaning and loss) and nursing staff. A relationship of both trust and time is needed to encourage relatives to use the QPL. As we found that relatives can use the QPL as an aid to discuss complaints, we found that nursing staff can perceive this as a threat, resulting in their reluctance to use the QPL. This points to an urgent need for good training of professionals, and perhaps continued training on the job that will allow them to feel confident enough to use the QPL.

## Figures and Tables

**Table 1 geriatrics-08-00023-t001:** Selected codes addressing the research questions in the interviews and fieldnotes.

Theme	Code
Atmosphere in which there is room to identify and address needs	Workload Hierarchy Interests; what is good for whom
Advance care planning: reactive and proactive care	Respond based on protocol Match need
Misunderstanding and conflict	Tensions in care relationship Unspoken reproach Discussion without confidence in outcome
Use, acceptance, and concerns around the QPL	Objections Suggestions for improvement Welcoming
Loss	Loss experiences of professionals and relatives Loss experiences of the person with dementia
Meaningful daily activities	Setting daily goals Activities that match personal interests

**Table 2 geriatrics-08-00023-t002:** The formal interviews during the ethnographic fieldwork.

	Interviews before the ACP Conversation and Duration	Interview after the ACP Conversation and Duration	Use of QPL
Duo interview with 2 relatives	Interview, 1 h, 40 min, 27 s	Interview, 1 h, 15 min, 1 s	Yes
Relative	Two interviews, 1 h, 10 min and 12 min	Interview, 32 min. 09 s.	Yes
Elderly care physician	Interview, 51 min. 19 s.		No
Nurse assistant	Interview, 34 min. 21 s.		No
Relative	Interview, 50 min.	Interview, 29 min. 13 s.	Yes
Nurse Bsc.	Interview, 51 min. 45 s.		No

ACP: advance care planning, QPL: question prompt list.

**Table 3 geriatrics-08-00023-t003:** Characteristics of persons interviewed in the ethnographic study.

Item	Relatives (4)	Professionals (3)
Age range	40–60 years	35–40 years
Relationship with resident	Kin (3), in-laws (1)	Elderly care physician (1), Nurse BSc (1), Nurse assistant (1)
Gender	All women	All women
Education level	Technical/trade school; mid-level (2), BSc level (2)	
Working hours	<16 (1), 32–40 (3)	24–32 h (all 3)
Religion	No religious affiliation	Catholic (1), spiritually connected (2)

## Data Availability

The following data are kept in the protected project folder of the Department of Public Health and Primary Care (PHEG), which only the researchers have access to: the questionnaires and the transcripts of the recordings. There are codes only and no names on any of these documents. After transcribing the interviews and checking for accuracy and quality, the recordings were deleted. All data and the code are stored in accordance with applicable laws and regulations. The data are stored for 15 years. No open access will be provided to the interview data because this might render participants identifiable.

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
