# Peer review of "Talking about Meaning and Loss with Relatives of Persons with Dementia: An Ethnographic Study in a Nursing Home"

_geriatrics, 2023, doi:10.3390/geriatrics8010023_

Round 1
Reviewer 1 Report
Talking about meaning and loss with relatives of persons with 37 dementia: An ethnographic study in the nursing home
The research aimed to
1.Evaluate inducing and enhancing conversations about meaning and loss in the context of nurse-led ACP conversations in dementia.
2.Evaluate the extent to which a question prompt list that includes a section about meaning and loss can meet the needs of relatives of nursing home residents with dementia and nursing staff.
It addresses important research questions in the field of palliative care and I congratulate the authors on the job well done. I have a couple of comments.
General comment: I recommend that the study is guided by a framework for conducting and reporting qualitative research for example the COREQ.
Other comments: The abstract should include the aim of the study.
Introduction: The authors cite extensive cancer related literature regarding communication. There is interesting literature on communication in dementia, I think it is more useful to cite this literature as opposed to the citing literature from cancer populations. Else the close relationship between the two conditions should be explained.
In table 2 please clarify if what we have in the brackets is the sample size (n=4), else it is unclear.
There are inconsistencies in the way quotes are referenced for example we have the type of respondent and the gender and at times only the type of respondent for example [ (Relative 3) and then ’ (Relative 2, woman). Then some for some we have respondents details on number and not for others if even the numerical ID is just one because we have one respondent it should be stated for consistency.
I felt more should have been said about the limitations, the fact that members of the research team had to engage in conflict management and resolution, how did this affect the study?
What biases could members of the research team have introduced?
For implications, looks like caregiver satisfaction with services can create tension and make it difficult to advance ACP despite the proposed innovation of enhanced conversations, what are the implications for this for practice and moving forward the best practice of ACP?
Reviewer 2 Report
Dear authors,
this is very interest and valuable manuscript. I submit my recommendation that I think will improve the manuscript.

Round 2
Reviewer 2 Report
Dear authors,
thanks for considering the recommendations.
I believe the manuscript is now improved and clearer and easier to follow for your valuable results.
I would also suggest that you provide explanations of abbreviations below the tables and that you move the Ethics paragraph to the end of the manuscript as requested by the journal template.
Best regard,
